# Challenges in Care for Non-COVID-19 Patients with Severe Chronic Illnesses during COVID-19 Pandemic: A Qualitative Study of Healthcare Providers Working around Acute Care Hospitals in South Korea

**DOI:** 10.3390/healthcare11040611

**Published:** 2023-02-17

**Authors:** Yejin Kim, Jeong Mi Shin, Shin Hye Yoo, Bhumsuk Keam

**Affiliations:** 1Center for Palliative Care and Clinical Ethics, Seoul National University Hospital, Seoul 03080, Republic of Korea; 2Public Healthcare Center, Seoul National University Hospital, Seoul 03080, Republic of Korea; 3Department of Internal Medicine, Seoul National University Hospital, Seoul 03080, Republic of Korea

**Keywords:** healthcare, quality, chronic disease, COVID-19, public health

## Abstract

Background: The COVID-19 epidemic has afflicted patients with severe chronic illnesses who need continuous care between home and hospitals. This qualitative study examines the experiences and challenges of healthcare providers around acute care hospitals who have cared for patients with severe chronic illness in non-COVID-19 situations during the pandemic. Methods: Eight healthcare providers, who work in various healthcare settings around acute care hospitals and frequently care for non-COVID-19 patients with severe chronic illnesses, were recruited using purposive sampling from September to October 2021 in South Korea. The interviews were subjected to thematic analysis. Results: Four overarching themes were identified: (1) deterioration in the quality of care at various settings; (2) new emerging systemic problems; (3) healthcare providers holding on but reaching their limit; and (4) a decline in the quality of life of patients at the end of their lives, and their caregivers. Conclusion: Healthcare providers of non-COVID-19 patients with severe chronic illnesses reported that the quality of care was declining due to the structural problems of the healthcare system and policies centered solely on the prevention and control of COVID-19. Systematic solutions are needed for appropriate and seamless care for non-infected patients with severe chronic illness in the pandemic.

## 1. Introduction

The coronavirus disease-19 (COVID-19) has led to excess deaths [1] and had a significant impact on access to healthcare resources and healthcare delivery [2]. It has also affected the use of healthcare services, and the clinical outcomes of patients with severe chronic illnesses [3,4], who frequently need continuous medical care and caregiving. During the pandemic, strategies for these patients should be both to reduce the risk of infectious disease and to maintain high quality of care, avoiding collateral damage from delayed treatment [5,6].

For the first two years of the COVID-19 pandemic, specifically before the Omicron variant displaced the Delta variant, South Korea has controlled the pandemic without resorting to a national lockdown [7] or reducing elective or acute care for patients with non-COVID-19 conditions, under the following quarantine strategies for COVID-19: aggressive screening and strict visitation policies. In healthcare institutions, the outdoor screening clinic and isolated facility of the emergency room have been utilized to isolate a person with COVID-19-suspicious symptoms until he/she is proven COVID-19 negative. Most hospitals limited the number of caregivers who could reside along with patients to one person and required caregivers to obtain a COVID-19 negative result. These countermeasures were deemed crucial to the success of quarantine. However, behind the success were the cessation of hospice services in public hospitals for operating designated hospitals for infectious diseases, prohibited visits in long-term-care hospitals (until March 2021), and limited compassionate visits, such as the time of death, in hospitals [8]. Moreover, the spread of COVID-19 has adversely affected the emergency medical system in South Korea [9].

Little has been explored about the impact of the COVID-19 pandemic and related quarantine measures on care for patients with severe chronic illness, especially in non-COVID-19 conditions, before and after treatment at acute care hospitals in South Korea. Patients with severe chronic illnesses suffering from non-COVID-19 conditions mainly access healthcare services at acute care hospitals. As their health condition worsens, they use emergency medical systems more frequently. Eventually, as staying at home without medical aid becomes difficult, they start using home care services, long-term-care facilities, and hospice care services [10,11]. The changes in healthcare delivery and policies at those healthcare institutions during the pandemic may make it difficult for healthcare providers (HCPs) to provide continuous care for patients with severe chronic illnesses in non-COVID-19 conditions [12,13].

There has been limited reporting on the challenges and experiences of caring for patients with severe chronic illnesses during the COVID-19 pandemic in South Korea. As the emerging pandemic situations may develop numerous impediments to substantial quantitative research due to social distancing policies or lockdowns, several studies [14,15,16] were undertaken for investigation and overall comprehension of multifaceted phenomena of healthcare utilization using qualitative research approaches. Given these similar studies, we believe that qualitative research that allows for flexible design, flexibility, and holistic accounting will be effective in this setting. Therefore, the aim of this study is to conduct exploratory qualitative research to identify the experiences and challenges of HCPs working around acute care hospitals who take care of patients with severe chronic illnesses in non-COVID-19 conditions.

## 2. Materials and Methods

### 2.1. Design

The purpose of this study is to look at unique phenomena that are shown in special areas around acute care hospitals in rare situations known as pandemics. A qualitative research method is useful when an exploratory approach to an unknown subject is needed to gain an understanding of the living experience from the perspective of people actually living a certain life [17]. Furthermore, qualitative research is context-dependent, allowing us to grasp how events, actions, and meanings are shaped by the unique settings in which they occur [18], so we suggest that the qualitative research approach is suitable since it is consistent with the goal of discovering new concerns during the pandemic that were not revealed in the setting of this study.

This study was conducted through the processes of research question development, interviews, and thematic analysis based on a generic qualitative research method without adopting a specific theory or approach, as it explored problems that have not yet been identified. Generic qualitative research avoids being strictly defined or buried in a single established methodology and is based on basic and common approaches that penetrate various approaches as a research tool that can effectively answer research questions, as compared to ontological and epistemological premises. The study was structured following the consolidated criteria for reporting qualitative research guidelines [19].

Two research questions are presented as follows: (1) How do HCPs working around acute care hospitals experience and perceive caring for non-COVID-19 patients with severe chronic illnesses during the COVID-19 pandemic, and what challenges do they face? And (2) What collateral damages are experienced by those patients with non-COVID-19 conditions and their caregivers as a result of such challenges? Prior to the period of this study, the non-Omicron variant was the most predominant variant of concern; therefore, this study’s research questions are limited to the pre-Omicron era of the COVID-19 pandemic.

### 2.2. Subjects

The sites according to the patient’s route of healthcare resource utilization are mentioned to clarify concerns regarding the site that non-COVID-19 patients with severe chronic illnesses visited prior to and after receiving treatment at an acute care hospital. On the list are an emergency department, a home, a long-term-care facility, and a hospice facility, in addition to an emergency delivery medical system to recruit HCPs at the site. Eight HCPs from an emergency room, emergency delivery medical system, home care, long-term-care facility, and hospice care facility working directly in the field during the COVID-19 pandemic were intentionally selected using purposive sampling. They were carefully chosen because they were thought to be best suited to answer research questions (Table 1). They (1) had worked at the site before the pandemic, (2) have more than 5 years of experience comparing pre- and post-pandemic settings, and (3) had worked at sites located in Seoul, the capital of South Korea. We chose Seoul as a setting because it has the most health and medical institutions per unit area in Korea, and all major hospitals are situated in Seoul, so the path of use of medical care around acute care hospitals is most clearly exposed.

### 2.3. Data Collection

In-depth interviews with the participants were conducted for about one month from 24 September 2021, at a place where they felt comfortable and safe. The questions used in the interviews were developed through collaboration between the two authors with experience in qualitative research (Table 2). The main interviewer worked as a doctor treating severely ill patients for eight years and has a doctoral degree in medicine. The secondary interviewer was a medical social worker who worked in the medical social work field for more than nine years and has a master’s degree in social welfare. The interviews were performed as semi-structured interviews using Fylan’s semi-structured interview guide [20], and open-ended questions were utilized to thoroughly investigate participants’ free expression. The main interviewer facilitated the interviews and the secondary interviewer checked whether any topics had been neglected and took notes on remarkable points from the participants’ discussion. All interviews were recorded and transcribed immediately after each interview. Moreover, notes from important conversations that occurred during the interviews were added to the transcripts. 

### 2.4. Data Analysis

The authors conducted a thematic analysis based on the grounded theory of Glaser [21] in order to undertake more systematic qualitative research. Grounded theory is a methodology typically employed for the development of theories, but it can also be applied to the thematic analysis of qualitative research. As we did not intend the development of theories in this study, we conducted subject analysis through the preceding stage of theory formation. The authors listened to the recorded interviews of the study participants and read the transcripts several times to grasp the overall content. According to Glaser’s theory [21], the authors identified the semantic unit using incident-by-incident coding, evaluated the pattern, conceptualized it, discovered categories and properties, and merged the themes through linkages between categories. These phases are not linear processes that occur simultaneously, but rather have experienced a cyclical process that alternately evaluates the preceding and following steps. 

## 3. Results

Four themes and 11 categories were revealed in the interview questions (Table 3).

### 3.1. Quality of Care Problems Arising in Various Settings

#### 3.1.1. Lack of Home-Based Medical Services for Patients with Severe Chronic Illnesses

The COVID-19 pandemic has highlighted the absence of stay-at-home medical services for patients with severe chronic illnesses. Due to the limited medical services available at home, the participants stated that patients with severe chronic illnesses at home were forced to visit the hospital for evaluation and treatment, though the process of medical treatment and transportation laid considerable physical, temporal, and economic burdens on these patients. Two participants (home care nurse, home care doctor) found that the number of severely ill patients choosing to stay at home had increased since the outbreak of COVID-19. They also observed that these patients endured their medical problems without adequate resolution, and ended up visiting the hospital, especially through the emergency room, only when their conditions grew worse.

“While the number of older adults and immobile people is increasing rapidly and more people want to stay at home due to COVID-19, healthcare services are not yet ready to provide care to patients at home. If severely ill patients want to stay at home, they should be able to do so. However, this choice is not available to them.” (Interviewee B)

“It is difficult for severely ill patients to wait at the hospital. Additionally, it is expensive to use vehicles in which they can lie down as they have to use a private ambulance and pay for the time it takes to see the doctor. Other expenses include they require suction while waiting to see the doctor, which requires a portable suction machine worth about KRW 800,000 and feeding tubes.” (Interviewee A)

“The tertiary hospitals tell the patients not to come back as there is little they can do for them. The patients are in enormous pain at home as the local hospitals are not allowed to prescribe narcotic painkillers to even cancer patients. When I ask the patients how they have been doing at home, they simply say that they have endured the illness. A patient with terminal pancreatic cancer used to visit local hospitals to receive fluids and painkillers. However, with growing difficulties in moving around, the patient is thinking of going to a long-term care facility or emergency room.” (Interviewee B)

#### 3.1.2. Increased Length of Stay Due to Procedural Delay in the Emergency Room

Two participants (emergency room nurse, emergency room transfer and referral) reported that the procedural duration for most non-COVID-19 patients with severe chronic illnesses had been prolonged in all processes in the emergency room as they commonly presented chronic symptoms shared by COVID-19 as well and were labeled as ‘suspected cases’. Furthermore, increased length of stay made severely ill patients spend most of their time—and some even died—in the emergency room.

“It is normal for patients with lung cancer to have fever and shortness of breath, but they are asked to take the COVID-19 screening each time they visit the hospital.” (Interviewee C)

“Although they undergo the examinations to receive medical treatment as they have no choice, the temporal and monetary burdens cause anxiety and difficulties among the patients.” (Interviewee D) 

“If patients have symptoms similar to those of COVID-19, they cannot receive general treatment and must be treated in the area reserved for patients suspected of having COVID-19. Once they are labeled as "suspected cases," they are destined to wait longer due to required pretreatment procedures. It will also take longer for a doctor or nurse to examine them, as the medical staff has to wear protective gear and come through the double doors of the negative-pressure isolation unit. Furthermore, some tests like the MRI or ultrasound, which are often required for cancer patients, cannot be performed unless the patients are first tested negative for COVID-19, and thus causes further delays in treatment.” (Interviewee C)

“There are times when some areas of the emergency room are so full of severely ill patients nearing the end of their lives that it feels like a hospice hospital. They cannot go to other hospitals, but are in need of treatment for their pain.” (Interviewee C)

#### 3.1.3. Deterioration in the Quality of End-of-Life Care for Inpatients

A participant (hospice coordinator) reported that strict visit restrictions during the pandemic made hospitalized patients with severe chronic illnesses nearing the end of their lives isolated from the people they were familiar with. The participant said that hospitalized patients felt that they had been deserted by their families, and they might suffer from anxiety and experience delirium frequently. Furthermore, as the visitation policies and the one resident caregiver rule applied to patients nearing death in hospice as well as to limitations in volunteer work and support programs, the participant said that there were no practical differences between hospice and non-hospice care, and in-hospice holistic care was curtailed.

“Human contact and direct face-to-face interaction are important, but the restrictions make this difficult. Patients often feel increasingly isolated and think, “My family has abandoned me here.” (…) Furthermore, people with severe delirium are more difficult to care for, and the frequent replacement of caregivers may make the delirium even worse.” (Interviewee G)

“With family caregivers or volunteers, the patients are well cared for, in aspects such as personal hygiene. Patients and caregivers expect hospice to have different advantages from general wards, but since hospice also needs to manage infection in the same way, the visitation policies and the one resident caregiver rule still apply. I feel sorry for the patient and caregiver when explaining this, and feel that the meaning of hospice has diminished a lot due to the restrictions on visitation.” (Interviewee G)

### 3.2. New Emerging Systemic Problems

#### 3.2.1. Severely Ill Patients in Need of Emergency Medical Care Unable to Use Emergency Rooms

Four participants (emergency room nurse, emergency room transfer and referral, emergency room doctor, and paramedic) observed that it became rather difficult for non-COVID-19 patients with severe chronic illnesses presenting any symptoms suspected of COVID-19 to access the emergency room during the pandemic. Since the emergency room did not have enough isolated room to accommodate all the patients with suspected COVID-19 symptoms, it was difficult for severely ill patients to find an isolated bed and delay every process in the transport chain. This was caused by a lack of prioritization in the allotment of emergency room beds, which operated on a first-come, first-served basis, without taking into account the severity and emergency of the cases. Often, patients gave up treatment as they failed to find an isolated bed or could not withstand the delayed transport process, leading to urgent and fatal situations involving the use of cardiopulmonary resuscitation. Additionally, when the emergency room was temporarily closed upon exceeding the capacity of emergency beds, a paradoxical situation arose where access to ambulances was blocked; however, access to patients walking in was not blocked, thereby threatening the safety of severely ill patients who attempted to enter the emergency room on foot.

“Emergency room treatments should be conducted in an isolated room if patients have any symptoms suspected of COVID-19. However, it is difficult to find an isolation room. Additionally, it is also difficult to transport patients to the hospital where they usually go. I called 40 emergency rooms for one patient, but they all said they could not accept the patient, and the patient had to be transported to a distant hospital 40 km away. (…). The news media also took notice of how the time it took for an ambulance to arrive at the scene, drive to the hospital, and take the patient home all increased. This is to be expected as the patients were being transported to distant hospitals and the time required for the processes increased.” (Interviewee F)

“About one-third of patients are concerned about mild symptoms or post-vaccination dyspnea, but these patients cannot be sent back. As a result, the admission of patients who may not be in a severe or acute state but are likely to be if not treated promptly is being delayed. There are frequent situations in which a patient who is at the threshold of risk with an oxygen saturation of about 94%, has no choice but to wait, as there is no bed.” (Interviewee D)

“Not long ago, a woman in her 20s, who needed plastic surgery treatment due to simple trauma, came to the emergency room saying that the waiting time for the outpatient treatment was too long. If such a patient enters the emergency room, a septic shock patient may not be admitted. (…) When the emergency room is closed due to too many patients, the route to 119 is blocked. However, there is no way to stop non-emergency patients from walking in. The case of a cardiac arrest patient who died after being rejected at 22 hospitals in Seoul, shows that if the scarce resources needed by severely ill patients are distributed to those with mild symptoms, the patients who really need them may not receive treatment. I do not know where all those heart failure and hematemesis patients are now. I suspect that these patients are unable to even come here.” (Interviewee E)

“There was a patient in an ambulance, whose oxygen saturation was barely 90%. While the ambulance was blocked as the emergency room was full, the patient constantly requested to enter the emergency room. The ambulance dropped the patient off near the emergency room and told the patient to walk, but as soon as the patient arrived at the emergency room, cardiopulmonary resuscitation had to be performed due to cardiac arrest.” (Interviewee C)

#### 3.2.2. Lack of Consistent and Realistic Guidelines

Two participants (emergency room doctor, paramedic) felt that the central guidelines from the Korea Centers for Disease Control and Prevention for COVID-19 precautions had been continuously conservative and demanding and did not consider the situation of non-COVID-19 patients with severe chronic illnesses. Due to the difficulty in following these central guidelines, each healthcare institution had individual guidelines, which were often ambiguous and lacked uniformity across institutions.

“If the standards of the Center for Disease Control are followed, then most of severely ill patients will not be able to access the emergency room. For example, patients with neutropenic fever or septic shock are not allowed to enter as per the guidelines. Hospitals with strict regulations do not accept patients on the grounds that they cannot rule out the possibility of COVID-19 infection when a medical history cannot be taken from patients due to a low level of consciousness.” (Interviewee E)

“It was hard to memorize all the different COVID-19 questionnaires for each hospital. “Did the patient come with a guardian?” “No, there is no guardian now.” “Then our hospital will not be able to accommodate the patient.” As such, patients may not be able to receive treatment without a guardian. (…) When the patient is under the influence of alcohol, they say that the patient’s words are not reliable, and the patient must be admitted to an isolated room. In general, patients need to be sent to an isolated room with a body temperature of 37.5 degrees, but some hospitals prefer to take preemptive measures by sending those with a body temperature of 37.3 degrees to an isolation room as well. There should be a unified standard for all hospitals to some extent, and there should be a clear ground for the screening of patients.” (Interviewee F)

### 3.3. Healthcare Providers Holding on but Reaching Their Limit

#### 3.3.1. Increased Workload Due to Infection Control Tasks

Three participants (paramedic, hospice coordinator, long-term-care hospital nurse) said that infection control procedures for screening and protecting non-COVID-19 severely ill patients with symptoms similar to those of COVID-19 led to temporal, physical, and psychological burdens on individual HCPs. The different guidelines of the institutions and the process of checking patient capacity had complicated the process even more. In addition, tasks, such as administrative tasks for infection control of patients, families, and paid caregivers, explanation of quarantine guidelines, and test confirmation, were added to the existing tasks of the staff with no reinforcement of the workforce. Time for counseling requested by the patients and families had increased, as had the burden of inquiries on the phone, which was rapidly increasing amid confusion due to quarantine guidelines or obstacles in the use of healthcare services.

“When it comes to quarantine management, it is difficult to breathe with a mask on. Putting on and taking off gloves and gowns also increases the workload. In addition, due to COVID-19, patient transfer and referral are not smooth, and the capacity of the emergency room is also reduced. The workload has also increased, even when working with severely ill patients.” (Interviewee H)

“Before COVID-19, I only contacted the emergency room in advance to see if it was possible to accommodate those who needed urgent treatment. Now, I must check all patients because of precaution measures. Before COVID-19, it took about 30 min to an hour to take care of one case. Now, it may take up to four to five hours for one case.” (Interviewee F)

“The COVID-19-related information of the guardians and caregivers is registered. This includes when and how they were tested for COVID-19, their relationships with the patient, and their contact information. This information should be entered into the computerized system and is being handled by nurses in the ward. (…) It takes a lot of time as nurses carry out the necessary verification work in relation to infection control and provide repeated explanation for the guardians and caregivers.” (Interviewee G)

#### 3.3.2. Experience of Internal and External Conflicts

Front-line HCPs were juggling the competing roles of controlling the spread of new infectious diseases and providing support to patients and families. Three participants (emergency room nurse, hospice coordinator, long-term-care hospital nurse) regarded the spread of infection as a threat and believed that it was necessary to control patients and families in accordance with the quarantine guidelines for visitation restrictions, tests, and transfers set by the government or healthcare institutions. However, the fatigue of HCPs had been building as it entailed persuading uncooperative patient caregivers or those with unreasonable demands. Moreover, they experienced internal conflict while imposing infection control rules on patients in pain at the end of their lives and on their families.

“Some people refuse to take the (COVID-19 screening) test and exchange caregivers against the rules and insist that an exception be made for their families. We have no choice but to talk about the principles. There is a dilemma in this situation and dealing with such cases is time-consuming. I have once even pleaded by saying, “If any one of our patients is tested positive for COVID-19, we have to close down the hospital and cannot take care of the patients at all.” (…) On the other hand, I was sad that I had to comply with the principles and stop families who had come from vast distances to see a dying relative.” (Interviewee G)

“In the COVID-19 situation, the relationship between the medical staff and the guardian sometimes becomes tense, as the medical staff have to say “No.” Guardians of severely ill patients are increasingly expressing their dissatisfaction with the medical staff, by saying, “This patient has been receiving treatment at this hospital. Why are you rejecting the patient and neglecting the patient in the emergency room?” Complaints such as these make the work of medical staff in the emergency room, who are already very exhausted due to the COVID-19 situation, even more difficult.” (Interviewee C)

“A caregiver leaving the premises is followed by a staff member. The staff supervises the caregiver to ensure that they go only for the intended purpose. From the perspective of caregivers, the staff may appear to be free to go whenever they wish. It is difficult to convince them as they had a lot of complaints.” (Interviewee H)

#### 3.3.3. Frustration and Burnout Due to Lack of Fundamental Solutions despite Individual Efforts

Two participants (emergency room doctor, paramedic) struggled to adapt to external circumstances beyond their individual control and felt a sense of loss and distress in a situation where they were blamed and held accountable for undesirable results. They were exhausted from resolving problems using temporary measures without a fundamental solution and felt a lack of a sense of achievement or reward in fulfilling their professional duties.

“The current situation is not at the discretion of the emergency room professor or head nurse. Prior to COVID-19, it was usually only necessary to reach an agreement between three people, to decide whether more patients could be admitted to the emergency room. Because of the risk of closing down the entire emergency room, the staff in charge cannot decide whether or not to accept a patient suspected of having COVID-19. It is necessary to consult with the in-hospital infection control team, the Disaster and Safety Countermeasures Headquarters, the Seoul Metropolitan Government, the Central Disaster Management Headquarters, and the Korea Centers for Disease Control and Prevention.” (Interviewee E)

“When there is a complaint that a patient was dead in an ambulance because the paramedics could not locate the hospital in time, those paramedics may be subjected to penalty or legal responsibility even when it may have been impossible to save the patient regardless of the will of the paramedics. Paramedics feel a great sense of loss and distress and may not want to keep working when they are criticized despite their sincere efforts.” (Interviewee F)

### 3.4. Decline in the Quality of Life of Patients at the End of Their Lives, and Their Caregivers

#### 3.4.1. Choosing Care Options That Might Never Have Otherwise been Chosen

Although severely ill patients and their families hoped to choose the “best care option” by considering the patient’s medical condition and the family’s capacity for caregiving, they were forced to choose the care options that avoided the worst-case scenario due to a decrease in the number of available healthcare institutions, strict caregiver-related policies, and lack of care resources such as care workers. For example, two participants (home care nurse, emergency room transfer and referral) felt that patients and their families did not choose hospitalization due to worries about possible infection related to group living even though it was the best option and endangered themselves while struggling with inadequate care, even with worsening symptoms.

“Each hospital has different standards and operating methods for admission, which have become more stringent since the COVID-19 outbreak. It is impossible to change shifts between caregivers, and the caregivers in charge are locked up in the hospital like prisoners. Therefore, our patients are choosing a small number of “rooms that do not require nursing care.” As a result, there are very few options. There are a lot of cancellations in the search for the available options. The number of hospitals that were consistently linked with our hospital was about 10 before COVID-19, but now it has decreased to less than 5.” (Interviewee D)

“There are patients who are hesitant to go to the hospital despite the need for inpatient care as it has become impossible to change shifts between caregivers at hospitals. Even during home care, the patient’s family members may be anxious about strangers coming to the house and decide to care for the patient themselves, even though they need professional help. The well-established care system has been broken by the COVID-19 pandemic. (…) I think the biggest problem is the restrictions on freedom; however, this problem is not explicit. Opportunities are limited, with a lot of difficulties, which may appear as a matter of personal choice from the outside, but in fact, there is no choice.” (Interviewee B)

#### 3.4.2. Notorious Burden of Patient Care on Family Caregivers

Unlike the flexible care options according to each family’s circumstances prior to COVID-19, it was found that the cases in which only a single family member oversaw providing patient care due to restricted duty shifts had increased since the pandemic. Two participants (home care doctor, hospice coordinator) observed that the ones who could not share the duty with other family members suffered from severe physical and emotional burnout in the absence of adequate social support systems and were more isolated and exhausted due to social distancing, even in hospice. With only one person in charge of caring, problems with livelihood commonly occurred, and conflicts arose between family members of patients, such as other family members not in charge of patient care criticizing the insufficiency of care. The burnout of the family caregivers led to a cascading decline in the quality of end-of-life care for patients.

”The one family member who takes full charge of patient care becomes completely exhausted. At first, the family member may have started it with good intentions, but there is no one else sharing the duty of patient care with them. In the past, family members could receive help from volunteers to perform physical care for patients, such as chaining their positions, or they could take a break while the volunteers bathed or massaged the patient. Without such support, family members in charge of patient care often suffer from physical burnout. Also, in the past, caregivers in similar circumstances exchanged support and had a positive influence on each other, but now they all live under a screen and feel emotionally isolated.” (Interviewee G)

“A patient’s son took a leave for patient care. Unlike the good intentions at the beginning of caring for the patient, his concerns about prolonged patient care were expressed in his words, "How much longer do you think the patient will live? I will lose my job if it gets any longer.” Although family members do not wish for the quick death of patients, they should also deal with the complex realities of their lives.” (Interviewee B)

“With only one person in charge of caring, conflicts arise between family members of patients. Despite the burnout of the family member in charge of patient care, other family members may criticize the insufficiency of care without being directly involved in patient care.” (Interviewee G)

#### 3.4.3. Death with Dignity Brushed Aside by Quarantine

A dignified and prepared end of life with loved ones had become hard to achieve due to the policy that prioritized the prevention of COVID-19 infection. Due to the strict visitation policies during hospitalization, patients tended to stay at home as much as possible, and often ended up dying at home without any preparation, leaving family members in anxiety and confusion. Three participants (home care nurse, home care doctor, long-term-care hospital nurse) felt that patients and their families suffered and wasted precious time in finding an emergency bed while dying in an ambulance, or being required to be tested to prove that they were ‘non-COVID-19’ to use funeral homes, instead of preparing for the moment of parting with dignity and comfort. Rituals and funerals to mourn the deceased had also become unconventional and restricted due to social distancing. Despite this situation, even the system to support high-risk bereaving families was not functioning.

“Patients may not want to spend the last hours of their lives with only one member of their family. Thus, there has been an increase in the number of chronically ill patients staying at home as much as possible (…).. Additionally, there have been cases in which patients were discharged with a plan to receive hospice care, but ended up dying at home while their admission to the hospice was postponed due to a variety of restrictions.” (Interviewee B)

“In one case, a dying patient at home was showing the symptoms of death rattle - shortness of breath and coughing, and it was painful for the caregivers to watch. The patient was not even able to drink water, and the family members asked whether the patient should be given medicine or taken to the hospital. It is necessary to provide proper education for family members to promote their understanding, and prepare them for the end-of-life process.” (Interviewee A)

“The XX funeral home only accepts people who have been tested negative for COVID-19. Those who wish to use the funeral home should be tested for COVID-19 in advance, even though they are dying.” (Interviewee H)

“I think the most difficult thing is that patients pass away without spending enough time with the people they love. Like a wedding, such a ritual is an important memory for people. Not having this ritual with the patient and not meeting the patient before they pass leaves a great regret among family members.” (Interviewee B)

## 4. Discussion

Before the Omicron era, South Korea’s quarantine policy was evaluated as successful, as it aimed not to limit medical care officially for non-COVID-19 patients. However, this study has revealed significant rifts in the care provided to non-COVID-19 patients with severe chronic illnesses, like overall problems in caring for these patients shown in previous studies of other countries experiencing the total collapse of the medical delivery system [2,12,22]. We suggest that non-COVID-19 patients with severe chronic illnesses can be adversely affected even in a situation with relatively few confirmed cases of COVID-19. This could lead to collateral damage by devastating the quality of life—and death—of patients with severe chronic illnesses and their caregivers [23].

The first theme revealed in this study, regarding the “quality of care problems arising in various settings,” is consistent with the results of the overseas study [12]. Home care HCPs were expected to fully support patients wanting to somehow stay at home [24,25]; however, they faced difficulties in keeping up with patient needs. Comparatively, non-COVID-19-related hospitalizations were reduced in the pandemic area compared to pre-pandemic levels worldwide [25,26,27]. Due to illness progression and complications, hospitalization is sometimes necessary for patients with severe chronic illnesses, and this decrease in hospitalization indicates that pandemics are causing collateral damage. More patients were seeking care outside of acute care hospitals but were not hospitalized; yet services outside of the hospital remained inadequate. This study also found that the psychological and practical barriers regarding the COVID-19 screening test, despite insurance coverage for it, led to patients at home experiencing inappropriate delays [22,25] in necessary treatment in two ways: (1) delayed emergency room visits and (2) a prolonged process in the emergency room. During hospitalization, the deteriorated quality of end-of-life care due to strict visitation policies has been reported in many countries [26,27,28], regardless of the severity and location of COVID-19, and South Korea has been no exception. Visitation restrictions deprive families of chances to bid proper farewells to patients before their death [12,23,29,30], and also disrupt the essence of hospices by not allowing patients and their loved ones to be together, which is the purpose of end-of-life care [13].

The second theme revealed that “emerging new problems compounded previous systemic problems.” Presently, healthcare completely revolves around COVID-19 [31,32,33], and the guidelines related to anti-disease care for patients with severe chronic illnesses are also only focused on reducing the risk of COVID-19 transmission. There has been a scarcity of practical guidelines or policies to care for non-COVID-19 patients with severe chronic illnesses [34]. This disrupts their healthcare delivery, leading to an ethical dilemma for HCPs [22,33].

The third theme, “HCPs holding on but reaching their limit,” suggests that HCPs are experiencing an increased burden of work, as reported by other studies abroad [29]. This, including education of patients and families regarding quarantine [24], hinders communication with patients and their families [29] and negatively affects the quality of life of HCPs and patient care; it is also associated with increased burnout. While more flexibility is required due to the uncertainty brought about by the COVID-19 pandemic [12], it has become difficult for individuals to respond flexibly to the situation due to strict quarantine guidelines. Meanwhile, HCPs experience conflict, frustration, and burnout due to criticism, ambiguous roles, and the scope of work exceeding their capabilities [12,13,29,35]. They feel caught in the middle of patients, their family caregivers, and the COVID-19 quarantine system.

Collateral damage, mainly revealed in our study, is recognized as a decrease in the quality of end-of-life care of severely ill patients in non-COVID-19 conditions and their informal caregivers during the pandemic. The patients are physically and psychologically isolated and spend the end of their lives in loneliness and helplessness [23,31,36]. Family members caring for patients struggled with patient care during the pandemic [23,30,37,38,39]. These challenges become more prominent as end-of-life approaches, as represented in the subtheme “Death with dignity brushed aside by quarantine.” Death during the pandemic can be traumatic for the caregiver, and causes prolonged grief disorder [36]; however, there is no systematic or institutional support for this aspect [28]. Interestingly, the distinctive subtheme of this study revealed that these care issues were regarded as matters of personal choice [24], even though patients and caregivers were given no other options [30].

## 5. Conclusions

This study has enabled investigating the overall flow and framework of care for non-COVID-19 patients with severe chronic illnesses by illuminating issues. This is expected to help prepare policies to maintain high-quality care for patients with severe non-infectious diseases during a pandemic. In addition, identifying the collateral damage to non-COVID-19 patients and their caregivers, when the intensity of the pandemic was not so severe in South Korea, has demonstrated the weak link in the system prior to a surge in new infections, enabling a preemptive response.

However, this study has several limitations. First, in this study, only eight participants were included, which may compromise the data saturation and limit the scope of the study. Second, this study has not included analysis of community HCPs and primary care providers, thus failing to cover all sectors of medical care in the community for patients with severe chronic illnesses. Third, interpretation of the results is limited by the possibility of a discrepancy between the collateral damage recognized by patients and their families and the collateral damage estimated by the perception of HCPs. Furthermore, the research questions of this study are limited to the pre-Omicron era, which may limit the interpretation of the study results and clinical implications during the COVID-19 pandemic.

In conclusion, HCPs have pointed out challenges in the COVID-19 pandemic, such as strict and rigid quarantine policies that did not consider the situation of patients with severe chronic illnesses in non-COVID-19 conditions and the medical system being shaken by the relatively low COVID-19 burden as compared to the other countries due to its vulnerability. These challenges led to collateral damage, such as difficulties in using appropriate medical care for patients with severe chronic illnesses; deterioration in the quality of care; burnout in HCPs caring for severely ill patients; and, in turn, a decline in the quality of end of life with dignity and comfort. Therefore, it is necessary to break away from a policy that only focuses on preventing the spread of COVID-19 and prepare a policy as well as a solid system to provide appropriate and seamless care for severely ill patients in non-COVID-19 conditions.

## Figures and Tables

**Table 1 healthcare-11-00611-t001:** Characteristics of study participants.

Interviewee	Sex/Age	Occupation	Institution	Role
A	Female/43	Nurse	Tertiary hospital	Home care nursing
B	Female/35	Doctor	Tertiary hospital	Home care
C	Female/40	Nurse	Tertiary hospital	Emergency room nursing
D	Male/41	Nurse	Tertiary hospital	Emergency room transfer and referral
E	Male/43	Doctor	Tertiary hospital	Emergency room care
F	Male/35	Paramedic	Fire station	Ambulance transport
G	Female/41	Nurse	Hospice care	Coordinator
H	Female/56	Nurse	Long-term-care hospital	Nursing

**Table 2 healthcare-11-00611-t002:** Questionnaire used in the interviews.

Question Type	Question(s)
Opening question	Please provide an introduction to your current job and field of work, as well as relevant experience and academic background.
Transition question	What do you think has changed the most in your workplace due to the COVID-19 pandemic?
Key questions	What challenges are you experiencing on a personal level due to the COVID-19 pandemic? What challenges are you facing at the level of your organization due to the COVID-19 pandemic? What kind of challenges do you think the entire healthcare system is experiencing due to the COVID-19 pandemic? What challenges do you think the severely ill patients and their families in non-COVID-19 conditions are experiencing due to the COVID-19 pandemic?
Ending question	Please share any stories that have not been covered in the interview so far but that you think are important.

**Table 3 healthcare-11-00611-t003:** Themes and categories based on the thematic analysis.

Themes	Categories
Quality of care problems arising in various settings	Lack of home-based medical services for patients with severe chronic illnesses Increased length of stay due to procedural delay in the emergency room Deterioration in the quality of end-of-life care for inpatients
New emerging systemic problems	Severely ill patients in need of emergency medical care unable to use the emergency rooms Lack of consistent and realistic guidelines
Healthcare providers holding on but reaching their limit	Increased workload due to infection control tasks Experience of internal and external conflicts Frustration and burnout due to lack of fundamental solutions despite individual efforts
Decline in the quality of life of patients at the end of their lives, and their caregivers	Choosing care options that may never have otherwise been chosen Notorious burden of patient care on family caregivers Death with dignity brushed aside by quarantine

## Data Availability

The data presented in this study are available in the manuscript.

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
