# Peer review of "Challenges in Care for Non-COVID-19 Patients with Severe Chronic Illnesses during COVID-19 Pandemic: A Qualitative Study of Healthcare Providers Working around Acute Care Hospitals in South Korea"

_healthcare, 2023, doi:10.3390/healthcare11040611_

Round 1

Reviewer 1 Report

Dear authors,

Congratulations for the article. The presented article brings contributions, through its results and discussion, to the identification of aspects of improvement in the organization of health care related to the changes instituted in health services due to the pandemic. However, given some weaknesses, namely methodological ones, I suggest a set of rectifications.

Title: should be more specific. “Health professionals in South Korea” is a very large population for a study of this size and of a qualitative nature. The context of these health professionals should be better specified. Even if they are from different work environments, they must have a common point. Example: health workers at a teaching hospital in South Korea. Or a geographic region... something that better encapsulates the specific context of the study.

Abstract:

The objective presented in the abstract is not identical (clearly) to the one presented in the introduction.

In the abstract and in the text, the term must be used uniformly and unequivocally (it is suggested to always use “healthcare providers”) and not use others that may confuse, in the English language, the type of participants in the study.

Introduction:

From lines 55-58, the cited reference does not seem to be adequate to the sentence presented, given the reality described, date of citation and content of the citation. It must be revised or justified.

The presented objective is not complete. It can be improved by referring, if possible: health professionals from a university hospital in South Korea: e, patients with serious non-COVID-19 chronic diseases… during the COVID-19 pandemic. That is, to standardize the terms used in the title with the objective presented in the abstract and in the introduction – including the defined research questions.

Method (greater fragility)

Design: Although they do not use a priori categories, in a theoretical model of framing the phenomenon (because it is new), they should mention what were the theoretical principles used (author and qualitative analysis method) to follow the research method and qualitative and thematic analysis . They refer in the data analysis (further on) to reference authors, who must be cited here.

Participants: only 8 participants, which may compromise the achievement of objectives and data saturation. It is also not clear if these participants are all from the same region of South Korea. Specify better. Clarify, as it is not understandable. Even if this fact does not compromise the pertinence of publishing this article, it is a limitation of the scope of the study results. The limitations of the study must be written in the limitations of the investigation, in the conclusions.

Data collection instrument: there is no mention of the type of interview guide, whether it was structured or semi-structured... closed or open... .

Data analysis (fragility): the authors cited in line 108 are indirect citations. The authors must be cited directly, with direct access to the guidelines for qualitative analysis mentioned by these authors. This is important, as it denotes the rigorous theoretical framework for the method.

Results (fragility):

The way the results are presented does not show the number of participants who support the creation of each category – the representativeness is not perceived. The participants who refer it or, at least, the number of participants who refer it must be placed.

The way it is presented, placing the transcription of an interview speech at the end of each category, suggests that the categories that emerge from speeches are, almost always, of only one participant. This suggests that the investigation is at an early stage of data collection – that is, data saturation has not yet been reached and it is ideal to continue to collect the opinion of more participants, including representatives of different types of health professionals. Despite this, the article's method can be considered “acceptable” as it is a contribution to understanding the phenomenon.

If the category created is extracted from the speech of only one participant, they cannot put the results in the plural (as they have, for example, in line 125, “Some HCPs found….” Or “Participants reported… when it is not understood, if it was one or more or how many said it). Or it could be an English translation problem…

On line 142, they put a transcription (between quotation marks) that they claim to be from 2 interviewees… Did 2 interviewees say exactly the same thing, using the same words?

On line 258, instead of “omitted”, put an ellipsis (…).

Finally, despite the reduced number of participants, the results found are of great interest and relevance.

Discussion: Good discussion, reasoned and cross-referenced with other significant authors.

Conclusions:

It could have a chapter, separate from the discussion, called conclusions. Limitations should be written here in the conclusions.

To the limitations already described, it added: type and number of participants in the study limit the scope of the study; the research questions of this study are limited to the pre-Omicron era; others…

References:

The vast majority of the 31 references are current and within the last 5 years.

Reference number 10 should be weighed against its usefulness, since it is from 2013 and far from the contextual reality studied (already mentioned in previous comments, in the introduction).

Is reference 13 on the qualitative method from 1965? There is a date error that must be revised with the correct reference – as its publication appears in the form of an article in 2014 (originally it was a book). It should be reviewed or it may be a better option to cite an author or a specific publication on the most current method.

Ethical principles:

The authors describe the use of informed consent and the number of approval by the responsible board and ethical principles seem to be fulfilled.

Reviewer 2 Report

In this study performed qualitative research to identify the experiences, challenges, and specific problems faced by healthcare providers(HCPs) taking care of patients with severe chronic illnesses. However, there are some issues regarding the quality of study presentation and some queries related to processes that should be addressed by the authors before any decision for publication as listed in the following.  

Introduction

1.   In the introduction a gap for similar studies that have assessed such healthcare for patients with severe chronic illnesses is felt. In fact providing a basis for other similar methods that have been used in a same way consistent with the aim of current study will help readers to understand the importance of such new strategies as an alternative for traditional methods. 

2.   It would be helpful if the provide specific grounds for the legitimacy of using the qualitative research.

Methods

3.   Did the research subjects agree to share their experiences? Or are you motivated?

4.   How did the discussion process for the selection of research questions take place?

Discussion

5.   Please provide the implications of the study to improve healthcare system in the conclusion. 

Round 2

Reviewer 1 Report

Dear authors,

Congratulations for the effort in corresponding to the suggestions that I made. The article has improved its clarity and quality.

Author Response

Thank you for your kind and thoughtful comments.